# Rotating Gantries Provide Individualized Beam Arrangements for Charged Particle Therapy

**DOI:** 10.3390/cancers15072044

**Published:** 2023-03-29

**Authors:** Siven Chinniah, Amanda J. Deisher, Michael G. Herman, Jedediah E. Johnson, Anita Mahajan, Robert L. Foote

**Affiliations:** 1Mayo Clinic Alix School of Medicine, Jacksonville, FL 32224, USA; 2Department of Radiation Oncology, Division of Medical Physics, Rochester, MN 55905, USA; 3Department of Radiation Oncology, Mayo Clinic, Rochester, MN 55905, USA

**Keywords:** charged particle therapy, rotating gantries, carbon ion radiotherapy

## Abstract

**Simple Summary:**

Carbon ion radiotherapy (CIRT) facilities are proliferating throughout Asia and Europe. They are costly to build, operate and maintain. Rotating gantries can deliver individualized treatments to patients with CIRT-eligible malignancies due to the increased number of choices of beam angles for treatment delivery. They help to minimize compromises in the treatment planning process, optimizing local tumor control and reducing the risk for acute and late adverse events, but they are expensive. Our aim is to retrospectively report the number and range of beam angles utilized to deliver proton therapy with a rotating gantry to a wide variety of patients with CIRT-eligible malignancies and to determine the proportion of tumor sites treated with unimodal, bimodal and multimodal groupings of beam angles. We found that only esophagus and pancreas cancers were treated with unimodal or bimodal beam groupings. Rotating gantries provide individualized beam arrangements for most CIRT-eligible patients.

**Abstract:**

Purpose: This study evaluates beam angles used to generate highly individualized proton therapy treatment plans for patients eligible for carbon ion radiotherapy (CIRT). Methods and Materials: We retrospectively evaluated patients treated with pencil beam scanning intensity modulated proton therapy from 2015 to 2020 who had indications for CIRT. Patients were treated with a 190° rotating gantry with a robotic patient positioning system. Treatment plans were individualized to provide maximal prescription dose delivery to the tumor target volume while sparing organs at risk. The utilized beam angles were grouped, and anatomic sites with at least 10 different beam angles were sorted into histograms. Results: A total of 467 patients with 484 plans and 1196 unique beam angles were evaluated and characterized by anatomic treatment site and the number of beam angles utilized. The most common beam angles used were 0° and 180°. A wide range of beam angles were used in treating almost all anatomic sites. Only esophageal cancers had a predominantly unimodal grouping of beam angles. Pancreas cancers showed a modest grouping of beam angles. Conclusions: The wide distribution of beam angles used to treat CIRT-eligible patients suggests that a rotating gantry is optimal to provide highly individualized beam arrangements.

## 1. Introduction

The use of charged particle therapy in cancer treatment has been growing in the United States and worldwide, with 38 centers in the United States and more than 110 worldwide and an additional 65 more under construction or in the planning phase globally [1,2,3]. The expansion of proton therapy (currently 99 centers) and carbon ion radiotherapy (CIRT, currently 13 centers) has required a significant financial investment, with the cost of a single-room proton therapy center estimated at about $20–40 million USD and multi-room or multi-ion centers costing on the order of $200 million USD [4]. One of the major contributors to the cost of establishing charged particle therapy centers is the gantry, which allows precise beam rotation to deliver highly individualized treatments from a wide array of angles. This cost is estimated to be $7 to $10 million USD for a proton gantry, with higher costs associated with heavier particles such as carbon ions (about $45 to $50 million USD).

However, the added cost and technical complexities in constructing and maintaining rotating gantries have limited their use, especially in CIRT centers [5,6]. As a result, of the 13 CIRT centers in the world currently in operation as of June 2022, only three utilize rotating gantries: the Heidelberg Ion Therapy Center in Heidelberg, Germany, which has a gantry weighing 600 tons, the QST Hospital in Chiba, Japan, which has a superconducting rotating gantry weighing about 300 tons, and the East Japan Heavy Ion Center at Yamagata University Hospital in Yamagata, Japan, with a superconducting rotating gantry weighing about 200 tons [7,8,9,10,11]. Two CIRT centers are currently in development in South Korea that will employ a combination of fixed beam and rotating gantry systems; one at Yonsei University Hospital in Seoul and one at Seoul National University Hospital in Busan, and these centers will employ the same gantry system used at Yamagata University Hospital [1].

With the costs and complexities involved in building a charged particle therapy center with a rotating gantry, the question of whether a rotating gantry is truly essential for treatment delivery is worth asking. In theory, the rotating gantry system, which is capable of a full 360 degrees of potential treatment angles, should allow for a lower dose to the organs at risk and more complete coverage of the treatment target with the prescription dose when compared to a limited number of fixed beam angles. Several studies have compared treatment plans between the rotating gantry and fixed beam systems for both proton therapy and CIRT. In a study evaluating the use of CIRT in pancreatic cancer, rotating gantry systems were found to have comparable or superior dose distributions in the surrounding normal tissues while achieving the same target coverage [6]. In a study of proton therapy planning for three patients with prostate cancer, plans with three angles (which would be best facilitated with a rotating gantry as opposed to multiple fixed beams) produced significant improvements in rectal sparing and “moderate improvements” in bladder sparing compared to those with either two lateral angles or two optimized angles [12].

In contrast, in a study assessing photon and CIRT in the treatment of five patients with skull base meningiomas, proton and CIRT plans using horizontal beams were able to achieve satisfactory dose distributions, with the use of a gantry being deemed as not essential in most cases [13]. This differs from previous work demonstrating the benefit of a gantry in obtaining optimal treatment angles for these tumors [14,15]. The more recent study conceded, however, that in scenarios where air-filled cavities must be avoided, rotating gantries may provide superior treatment plans and patient outcomes due to the availability of more beam angle options. A retrospective analysis of gantry-based proton treatments was published in 2016 by the Massachusetts General Hospital (MGH) and concluded that most beam approaches could be realized in a non-gantry system of delivery [14]. It should be noted there is variability in treatment modality, planning approaches, machine capabilities and changes over time and between institutions in the philosophy of tumor prescription dose coverage and dose constraints for organs at risk.

Additional advantages to a gantry-based treatment system include less deformation of organs during treatment planning imaging, image-guided treatment and treatment delivery, improved efficiency secondary to decreased patient positioning time, improved patient comfort secondary to patient position (no tilting or rotation of the treatment table) and a shorter total treatment duration time.

Herein we report on our retrospective evaluation of 484 unique treatment plans as well as 18 additional boost plans, focusing on the number and range of beam angles utilized in CIRT-eligible patients treated with pencil beam scanning (PBS) intensity modulated proton therapy (IMPT). All plans in this analysis were utilized in the delivery of treatment with rotating gantries at Mayo Clinic in Rochester, Minnesota, over a five-year period. Our aim is to report the number and range of beam angles utilized to deliver PBS IMPT to a wide variety of patients with CIRT-eligible malignancies and to determine the proportion of tumor sites treated with unimodal, bimodal and multimodal groupings of beam angles.

## 2. Patients and Methods

This retrospective chart review of previously treated patients was approved by the Mayo Clinic Institutional Review Board, and was deemed minimal risk. We initially screened patients who had been treated with pencil beam scanning intensity modulated proton therapy (IMPT) and simultaneous integrated boost technique between 22 June 2015 and 31 May 2020. Within this pool of patients, we further defined CIRT-eligible subjects as patients whose cancer characteristics (anatomic primary tumor site, histology, stage, lymph node involvement) have been reported to be successfully treated using CIRT [16]. Patients with breast cancer were included because of the ability to spare the heart, lungs and esophagus, and because of an interest in leveraging the higher LET of CIRT to treat locally advanced diseases, such as patients with poor tumor responses to chemotherapy, as is the case in some patients with triple-negative breast cancer. In addition, breast cancers with homologous recombination defects are more sensitive to higher LET than surrounding normal tissues. Also, the potential of treating the primary tumor and/or metastases to generate a potentially more robust immune response in combination with other immune-based therapies is of interest. Finally, patients with unresectable locally advanced primary breast cancer or recurrent previously irradiated breast cancer are situations where CIRT may be able to provide locoregional control and prolong survival. All patients would have been referred for CIRT if it were available within the United States. Some patients were referred for CIRT but were unable or unwilling to travel to foreign countries for treatment. Since CIRT was not an option for these patients, they were treated with proton therapy to take advantage of the Bragg peak sparing of organs at risk, the slight biologic advantage and hypofractionation. Because of the cost of charged particle therapy, our interest was in curative rather than palliative treatment; therefore, patients with metastatic cancer were excluded from the patient population analyzed. This study is based on the local, regional, national and international population of patients seen and treated with proton therapy at Mayo Clinic in Rochester, Minnesota.

Primary treatment plans, including some patients with more than one primary tumor being treated, and separate boost plans were included in this analysis. These plans were grouped based on the anatomic location of the primary tumor treatment site and the type of plan. The beam angles utilized for these subgroups were collected electronically from the treatment planning system (ARIA, Eclipse, Varian Medical Systems, Palo Alto, CA, USA).

All patients were treated with a 190° rotating gantry with a six degree of freedom robotic patient positioning system capable of rotating greater than 180° to facilitate both left- and right-sided beam angles (Hitachi, ProBeat V, Tokyo, Japan) (Figure 1). The image guidance system allows for patient positioning corrections of up to 3° in pitch, yaw and roll. Beam angles were grouped in bins of 5°, ranging from −5° to 185° (0° corresponding to an anterior beam in a supine patient orientation), and sorted into histograms for each anatomic site treated with at least 10 patient treatment plans.

The proton treatment plans were individualized to deliver ≥95% of the prescription dose to ≥95% of the target volume while limiting the dose to surrounding organs at risk according to evidence-based, consensus-driven, standard dose constraints, or lower, if possible. Areas of high biological dose (≥110% of the prescription physical dose) and high LET were kept within the tumor target volume and out of the adjacent organs at risk, as is done with CIRT treatment planning. The treatment plans were prescribed and approved by 25 different physicians representing a variety of treatment planning preferences. Our institution utilizes an in-house developed Monte Carlo dose calculation platform which complements the analytic pencil beam algorithm of the primary treatment planning system [17]. Using proton linear energy transfer (LET) distributions computed by the Monte Carlo system, we have implemented a biological dose model which accounts for the increased relative biological effect (RBE) at the end of the proton beam range [18].

Multiple planning iterations are requested of the certified medical dosimetrist and medical physicist by the responsible treating radiation oncologist prior to final plan acceptance and approval to individualize and optimize the treatment plan in terms of highest dose possible to as much of the tumor target volume as possible and as low a dose as possible to the smallest volume possible of the organs at risk. Each patient’s plan started with a first iteration using a lower number of beam angles and/or more standard beam angles (anterior, posterior, right and left lateral, 45° anterior or posterior oblique). In the iterative treatment planning process between the dosimetrist, physicist and physician, the beam angles were modified and the number of beams increased to optimize target coverage, minimize organ at risk dose including lowering biological dose and improve plan robustness by adding beams and adjusting beam angles, as is done with CIRT treatment planning. The plans included in this study represent what the treating physician determined to be the best clinical plan for the patient for optimal target coverage and sparing of organs at risk considering physical and biological dose [19,20,21,22,23,24,25,26,27,28,29]. Using fewer beams or more standard beam angles resulted in inferior treatment plans which were not accepted by the treating physician. The more beam angles and degrees of freedom used, the better for creating optimized biological dose distributions.

## 3. Results

A total of 467 patients with 484 individualized primary curative treatment plans were identified and included in this analysis. Plans were characterized by anatomic primary tumor treatment site, number of treatment beams utilized and range of treatment beam angles (Table 1). Of note, the treatment plans included 1196 unique treatment beams. Patients with treatment sites of the left and right breast, esophagus and liver had the most unique treatment beams. For all patients included in this study, 0° (direct anterior beam) and 180° (direct posterior beam) were the most common beam angles used, compromising a total of 201 treatment beams. Clusters of beam angles between anterior 0°, horizontal 90° and posterior 180° were utilized (Figure 2 and Figure 3). Plans still exhibited a wide range of treatment angles even after excluding the most common beam angles of 0° and 180° (Table 2). Figure 2 and Figure 3 demonstrate the number of beams used at each angle by primary tumor treatment site, demonstrating the variety of treatment sites treated with each beam angle.

Treatment plans utilized a median of 2.0 beam angles per anatomic primary tumor treatment site.

Table 3 depicts the number of beam angles used per patient with tumors located at each anatomic primary tumor treatment site. Each anatomic primary tumor treatment site had a median of two to three beam angles used per patient, except for uterine cancer, in which a single patient was treated with two beams using a single beam angle (140°), rotating the treatment couch 180° to treat the patient from the right and left side. Of the 467 total patients, 38 (8.1%) were treated with a single beam angle, 291 (62.3%) were treated with two beam angles, and 138 patients (29.6%) were treated with three or more beam angles.

Patients treated with combinations of standard fixed beamline system orientations of direct anterior or posterior (0° and 180°), horizontal (90°) or anterior or posterior oblique (45° and 135°) were assessed. Of the patients treated with a single beam angle, 92.3% were treated with a non-standard fixed beam angle. For patients treated with two beam angles, 89.1% were treated with combinations including one or more non-standard fixed beam angles. All patients treated with three or more beam angles were treated with combinations including one or more non-standard fixed beam angles.

Histograms were generated for primary curative plans for the anatomic primary tumor treatment sites utilizing the most treatment beams. Almost every site had a wide range of beam angles used in treatment delivery (Figure 3a). Only cancers of the esophagus were shown to have a strong unimodal grouping of beam angles, with 84% of the 112 beam angles fitting in the 160° ± 10° range, while cancers of the pancreas showed moderate grouping in beam angles, with 94% of the 42 beam angles fitting in either the 155° ± 10° range or 180° (Figure 3b). Histograms generated from the boost plans showed no central tendency of beam angles for any treatment site.

## 4. Discussion

The distribution of beam angles among the CIRT-eligible patients treated with proton therapy at our institution suggests that a rotating gantry can achieve highly individualized dose distributions. Of the cancer primary anatomic treatment sites and types examined in this study, only esophageal and pancreatic cancers had beam distributions suggesting the potential for treatment with two fixed beams. Esophageal cancers are typically treated with two oblique posterior beams, one from the left and one from the right, with the patient being rotated 180° by the patient positioning system to allow for left- and right-sided delivery to optimize sparing of the heart, lungs, liver and kidneys [30,31]. This would explain the small number of angles used in the treatment of this subset of cancers. For a fixed beam system, this would require tilting the patient, leading to discomfort, organ deformation and treatment planning complexities.

Alternatives to a rotating gantry system that allow for a greater variety of treatment beam angles in CIRT delivery are currently being explored because of the cost of the rotating gantry [32]. The use of a rotating upright chair with a single horizontal fixed beam delivery system has been studied in a variety of clinical settings to treat patients with cancers of the head and neck region and thorax, especially patients with physical limitations making supine treatment difficult, such as phrenic nerve damage or treatment couch weight limitations [33]. In treating head and neck cancers in a seated position, intrafraction and interfraction displacement were comparable to treatment in a supine position, and equivalent reproducibility was found between patients in each of these systems [34]. In palliative irradiation of lung cancers, treatment in the seated position allowed for increased lung volumes and thus a smaller percentage of healthy lung parenchyma exposed to radiation when compared to treatment in the supine position, with increased patient comfort in the seated position as well [35]. The seated system may provide an efficient, low-cost, 360-degree treatment beam angle alternative to a rotating gantry [36].

However, challenges exist in the variety of tumor sites that can be treated in the sitting or standing position including immobilization, motion management, treatment planning and image guidance during treatment. For instance, patients may not be able to keep still in the seated or standing position due to increased fatigability as a result of the cancer, treatment side effects or comorbid illnesses [34]. Even with a rotating gantry with the patient in a supine position, adequate immobilization is critical for ensuring the delivery of planned dose distributions and limiting toxicities to organs at risk [37].

In addition, imaging used for treatment planning and image-guided radiotherapy would need to be performed with vertical scanners with the patient in the seated or standing treatment position. Vertical CT systems have been developed and incorporated into radiotherapy treatment planning and delivery but are still uncommon. Vertical MRI scanners exist for orthopedic indications but have not been incorporated consistently into radiotherapy treatment planning and delivery [38,39]. Adequate imaging of the abdomen and pelvis may not be possible, thus limiting the number of patients eligible for CIRT with these systems. Currently, clinical data is not available to indicate what proportion of charged particle therapy patients can be treated in the seated position in a rotating chair with a single fixed beam.

The differences in results between our study and the prior MGH study may be due to differences in treatment planning approaches and philosophy regarding tumor target prescription dose coverage, treatment positioning, pencil beam scanning IMPT vs. passive scattering delivery and dose constraints for normal organs between the respective institutions [14]. Most notably, most of the treatment plans which the MGH study stated could be performed using a fixed beam system involved the patient in the sitting position or in both the sitting and supine positions. In addition, assumptions were made in the study based on technology that is not yet available, including the potential to treat a patient in both the sitting and supine position with the same treatment plan. These approaches may be complicated by relative organ displacements between these positions and patient comfort issues. The MGH study acknowledges the potential differences in movement, including respiration, between the sitting and supine positions that would need to be addressed to best treat patients with a variety of primary cancer anatomic treatment sites using a single fixed beam and a rotating chair.

Advancements in treatment planning techniques are another possible explanation for differing conclusions between this work and previous investigations. The capability of visualizing the effects of proton LET distribution and RBE through our Monte Carlo dose calculation system has allowed us to refine our planning techniques with the goal of minimizing and/or shifting the location of high biological dose regions. These approaches have driven the choice of beam angles and an increase in the number of beam angles delivered per plan, as a larger number of beams and angles allows for end-of-range effects to be manipulated and spread out. This is an important consideration for optimizing CIRT. Proton end-of-range effects have long been known in the particle therapy community, with mitigation strategies based on clinical experience and intuition. In the era of spot scanning and intensity-modulated proton therapy, however, it is more difficult to anticipate the distribution of high RBE dose regions. The combination of these factors and innovations has resulted in planning techniques at our institution that are not directly comparable to those employed in previous studies at other particle therapy centers. Prior investigations of fixed beam and gantry treatment plans did not have this treatment planning capability and may have come to a different conclusion based on biological dose and LET distribution considerations.

All but three of the current CIRT treatment facilities use a combination of fixed beams (horizontal, vertical and/or 45°) which may result in more compromises in the treatment planning process or an inability to treat all eligible patients [1]. This limitation in beam angles introduces inefficiencies in patient throughput and results in more patient discomfort from longer treatment times in the immobilization device on the treatment couch due to tilting the patient, multiple setups, and additional imaging needed for positioning and treatment delivery. For example, some treatments use a table tilt or roll to obtain more ideal angles. However, this requires several adaptive treatment plans to be created in response to anatomical deformation and resulting organ movement due to the uncertainties in dose distribution, organ sparing and target coverage, which results in compounding inefficiencies and costs [40]. In addition, the table tilt requires an additional 5–10 min of re-setup time compared to 1–2 min of gantry rotation.

Currently, proton gantries rotate at the same speed as conventional photon gantries, one revolution per minute. Current carbon ion gantries are slower at 0.5 revolution per minute. Both gantry systems provide the flexibility of 360 treatment beam angles. A 3-treatment beam angle head and neck patient can be treated in less than 10 min. Intensity-modulated particle therapy is not possible with fixed treatment beams because the treatment planning images are not the same for each treatment beam angle. Finally, treatment beam angles cannot be optimized on treatment planning CT images in a fixed treatment beam system.

One limitation of this study is the use of proton therapy-based plans as a proxy for CIRT. We are assuming the treatment beam angles selected for the accepted IMPT plan would be the same as an accepted intensity-modulated carbon ion radiotherapy plan. Although the dose deposition properties of the proton and carbon ion beams are similar in many ways, there are differences that need to be considered during the radiation treatment planning process. The entrance dose of carbon ion beams is generally lower than proton beams for an equivalent biological dose in a target volume. The lateral penumbra of carbon ion beams has a smaller sigma than protons, particularly at deeper patient depths where the effects of multiple coulomb scattering have broadened the proton beam penumbra.

The decreased entrance dose and sharper lateral penumbra of carbon ion beams are notable but provide only modest dosimetric advantages compared to protons; as such, we expect carbon ion beams to assist with achieving optimization goals but not to directly influence beam angles in some cases [41]. Carbon ion beams have a higher LET than protons, especially at the end of the range where biological models are needed to account for this increased biological damage, which was done for our proton treatment planning. In previous treatment planning studies comparing CIRT and proton therapy in various sites, planned treatment beam angles were the same between the two modalities [41,42,43,44].

As mentioned earlier, our proton practice has incorporated a custom LET evaluation and biological dose model into our dose optimization procedure, which makes our proton planning process more like that of carbon ion planning and differentiates this study from previous work. Finally, carbon ion beams exhibit a “fragmentation tail,” which increases the dose distal to the Bragg peak by a small, but non-negligible amount. The amplitude of this fragmentation tail depends on the specific characteristics of the carbon ion beam but is generally on the order of ~20% of the target dose. Doses on this scale are typically not high enough to approach the clinical tolerances of nearby organs at risk, and thus we postulate that the carbon ion fragmentation tail will not be a primary influence on treatment beam angle selection for most sites. Exceptions to this are possible, for example in the case of skull base planning, in which tumors may abut ocular structures, cranial nerves, the brainstem and the cerebrum, and breast planning, in which low radiation doses to the heart have been correlated with increased late toxicities [13,45]. Even in the unlikely event that these sites do not require a gantry to facilitate the best combination of treatment beam angles, the overarching conclusion of this analysis is that a rotating gantry can individualize treatment for a wide variety of primary anatomic tumor treatment sites. Additionally, with only three carbon ion gantry facilities in clinical operation, the majority of carbon ion planning experience has been influenced by the constraints of standard fixed beam angle geometries. This is a possible explanation for how nominal carbon ion planning techniques have evolved to generally utilize fewer treatment beam angles. A more widespread proliferation of carbon ion gantries could result in treatment planning techniques that more closely resemble those of photon and proton radiotherapy, where gantry-based systems have been the standard for many years [7].

Another limitation of the study is the patient population which may differ from one carbon ion radiotherapy facility to another, thus changing the proportion of patients who could be treated with a fixed beamline system and standard angles. In the future, with new technology for vertical imaging, immobilization, treatment planning and image guidance for 360° rotating treatment chairs or stands, it may be possible for more patients to be treated with a single fixed beamline at a standard angle; however, in the absence of clinical data demonstrating the feasibility and utility of this technology, at this time the rotating gantry system provides more flexibility in developing safe and effective individualized treatment plans for maximal target volume prescription dose coverage and reduced dose to organs at risk.

Based on the treatment planning work at our institution with a variety of particle therapy vendors, a CIRT system composed of three fixed beamlines would cost more than a rotating gantry system due to the number of steering and bending magnets required for three fixed beamlines compared to a single beamline for a rotating gantry. The rotating gantry would provide many more treatment beam angles allowing for more individualized treatment plans, more efficient treatment and more patient comfort.

## 5. Conclusions

Our retrospective analysis of CIRT-eligible patient IMPT treatment plans generated at our institution demonstrates the need for a wide variety of treatment angles for highly individualized radiotherapy delivery. These findings suggest that a rotating gantry system would be optimal to efficiently achieve individualized dose distributions with CIRT with the current available technology.

## Figures and Tables

**Figure 1 cancers-15-02044-f001:**
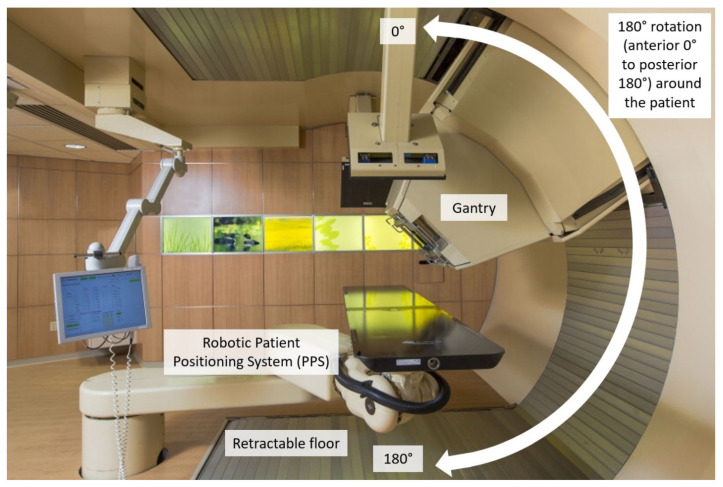
Photograph of the proton therapy treatment delivery system which consisted of a 190° rotating gantry with a six-degree-of-freedom robotic patient positioning system capable of rotating greater than 180° to facilitate both left- and right-sided beam angles (Hitachi, ProBeat V, Tokyo, Japan). For treatment planning and delivery, 360 beam angles are available to choose from. A rotation of 0° corresponds to an anterior beam in a supine patient orientation. A rotation of 180° corresponds to a posterior beam in a supine patient orientation.

**Figure 2 cancers-15-02044-f002:**
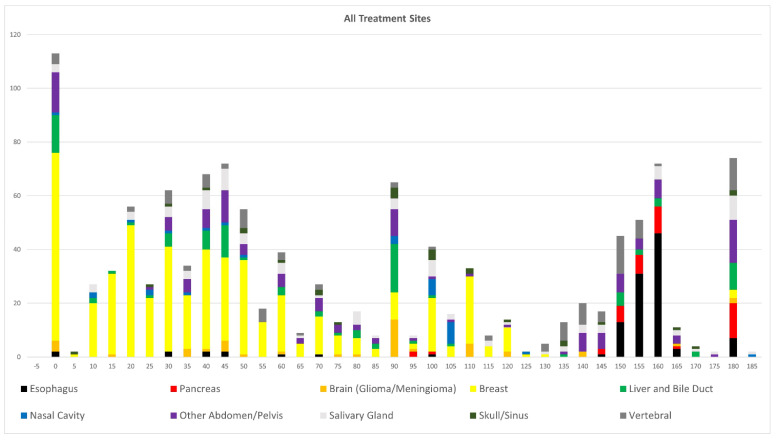
Demonstration of the number of beams (*y*-axis) used at each angle (*x*-axis) by primary tumor treatment site (color coded).

**Figure 3 cancers-15-02044-f003:**
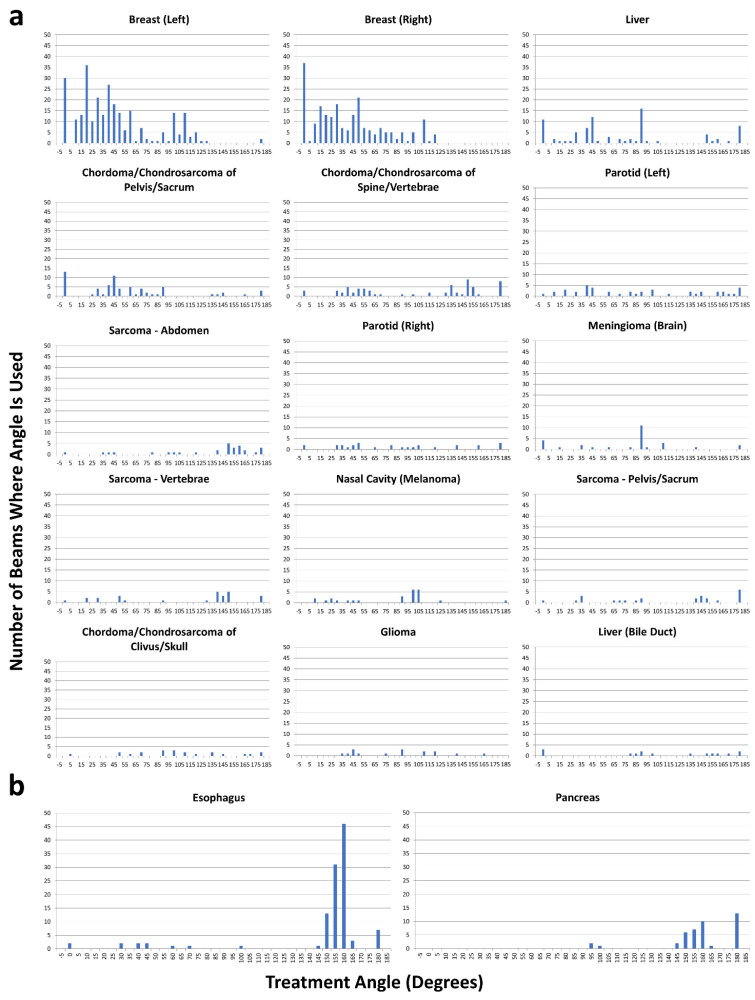
Histograms demonstrating the number of beams (*y*-axis) used at each angle (*x*-axis) by primary tumor treatment site. (**a**) Histograms generated for primary curative treatment plans for the anatomic primary tumor treatment sites utilizing the most treatment beams and having a multimodal distribution. (**b**) Histograms generated for esophagus and pancreas primary curative treatment plans. Only cancers of the esophagus were shown to have a predominantly unimodal grouping of beam angles, with 84% of the 112 beam angles fitting in the 160° ± 10° range, while cancers of the pancreas showed moderate bimodal grouping in beam angles, with 94% of the 42 beam angles fitting in either the 155° ± 10° range or 180°.

**Table 1 cancers-15-02044-t001:** Patients treated with proton therapy by treatment site.

	Patients	Initial Plans	Number of Beams	Range of Angles (°)
**Total**	467	484	1196	(0,185)
Adenoid Cystic Carcinoma (Nasal Cavity)	1	1	4	(0,105)
Adenoid Cystic Carcinoma (Oral Cavity)	1	1	3	(60,115)
Breast (Bilateral)	2	2	9	(0,180)
Breast (Left)	128	128	276	(0,180)
Breast (Right)	102	102	217	(0,120)
Chordoma/Chondrosarcoma of Chest Wall	1	1	3	(135,180)
Chordoma/Chondrosarcoma of Clivus/Skull	6	6	22	(5,180)
Chordoma/Chondrosarcoma of Pelvis/Sacrum	24	24	67	(0,180)
Chordoma/Chondrosarcoma of Spine/Vertebrae	17	20	66	(0,180)
Esophagus	54	55	120	(0,180)
Glioma	5	5	16	(35,165)
Kidney	5	5	12	(90,180)
Liver	28	32	84	(0,180)
Liver (Bile Duct)	6	6	15	(0,180)
Meningioma (Brain)	9	9	28	(0,180)
Nasal Cavity (Melanoma)	8	8	26	(10,185)
Pancreas	19	19	45	(95,180)
Parotid (Left)	13	14	44	(0,180)
Parotid (Right)	7	9	29	(0,180)
Sarcoma—Abdomen	11	11	30	(0,180)
Sarcoma—Pelvis/Sacrum	8	9	26	(0,180)
Sarcoma—Sinus/Skull	2	2	6	(25,100)
Sarcoma—Thorax	1	1	3	(70,155)
Sarcoma—Vertebrae	7	8	27	(0,180)
Submandibular Gland (Left)	4	4	13	(10,185)
Submandibular Gland (Right)	1	1	3	(45,180)
Uterus	1	1	2	(140,140)

**Table 2 cancers-15-02044-t002:** Patients treated with proton therapy by treatment site (excluding 0° and 180°).

	Number of Beams	Range of Angles (°)
**Total**	995	(5,185)
Adenoid Cystic Carcinoma (Nasal Cavity)	3	(35,105)
Adenoid Cystic Carcinoma (Oral Cavity)	3	(60,115)
Breast (Bilateral)	5	(40,100)
Breast (Left)	244	(10,130)
Breast (Right)	180	(5,120)
Chordoma/Chondrosarcoma of Chest Wall	2	(135,155)
Chordoma/Chondrosarcoma of Clivus/Skull	20	(5,170)
Chordoma/Chondrosarcoma of Pelvis/Sacrum	51	(25,165)
Chordoma/Chondrosarcoma of Spine/Vertebrae	55	(30,160)
Esophagus	103	(30,165)
Glioma	16	(35,165)
Kidney	8	(90,160)
Liver	65	(10,170)
Liver (Bile Duct)	10	(80,170)
Meningioma (Brain)	22	(15,140)
Nasal Cavity (Melanoma)	26	(10,185)
Pancreas	29	(95,165)
Parotid (Left)	39	(10,175)
Parotid (Right)	23	(30,160)
Sarcoma—Abdomen	25	(35,175)
Sarcoma—Pelvis/Sacrum	18	(30,160)
Sarcoma—Sinus/Skull	6	(25,100)
Sarcoma—Thorax	3	(70,155)
Sarcoma—Vertebrae	23	(20,150)
Submandibular Gland (Left)	12	(10,185)
Submandibular Gland (Right)	2	(45,45)
Uterus	2	(140,140)

**Table 3 cancers-15-02044-t003:** Median number of beam angles per patient by treatment site.

	Median Number of Beam Angles Per Patient
**All Sites**	2
Adenoid Cystic Carcinoma (Nasal Cavity)	3
Adenoid Cystic Carcinoma (Oral Cavity)	3
Breast (Bilateral)	3
Breast (Left)	2
Breast (Right)	2
Chordoma/Chondrosarcoma of Chest Wall	3
Chordoma/Chondrosarcoma of Clivus/Skull	3
Chordoma/Chondrosarcoma of Pelvis/Sacrum	2
Chordoma/Chondrosarcoma of Spine/Vertebrae	2
Esophagus	2
Glioma	3
Kidney	2
Liver	3
Liver (Bile Duct)	2.5
Meningioma (Brain)	3
Nasal Cavity (Melanoma)	2.5
Pancreas	2
Parotid (Left)	3
Parotid (Right)	3
Sarcoma—Abdomen	3
Sarcoma—Pelvis/Sacrum	2.5
Sarcoma—Sinus/Skull	3
Sarcoma—Thorax	3
Sarcoma—Vertebrae	2
Submandibular Gland (Left)	3
Submandibular Gland (Right)	2
Uterus	1

## Data Availability

Data will be made available to researchers who contact the corresponding authors and provide a methodologically sound proposal.

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
