# Peer review of "Rotating Gantries Provide Individualized Beam Arrangements for Charged Particle Therapy"

_cancers, 2023, doi:10.3390/cancers15072044_

Round 1

Reviewer 1 Report

Summary: The goal of this study is to determine the importance of a rotating gantry allowing for beam angle selection for future patients who may be treated with carbon ion radiotherapy. The authors performed a retrospective analysis of beam angle selection for patients treated with proton radiotherapy at a single institution in order to infer the importance of multiple beam angles in a real-world cohort of patients treated with chaged particle therapy. The major strength of this study is that it provides information on beam angle selection from a large number of patients actually treated with charged particle therapy in a real-world setting at a center where a rotating gantry is available, and suggests there is a perceived benefit from having a choice of multiple angles for treatment of different cancers. However, the study can not and does not directly address the question of the actual clinical importance of multiple beam angles for carbon ion therapy and the use of a rotating gantry to achieve this. The authors present a well written manuscript, but it is difficult to see the importance of the data.

General comments:

Although not stated, the inferred hypothesis of this study is that a range of beam angles provided by a rotating gantry is optimal for delivering carbon ion therapy. Unfortunately, the study design does not allow for testing of this hypothesis. A better design would be a dosimetric study comparing carbon therapy plans utilizing a limited number of fixed beams with plans utilizing optimal beam selection. Since there will almost certainly be a dosimetric benefit to a larger selection of beam angles, it would be important to further determine if there is a clinically significant difference in exposure of OARs. To keep to the spirit of the real-world conditions captured by this current study, a comparison could be made between carbon plans utilizing the same beam angles selected for proton plans and comparing them to plans utilizing a more limited fixed set of angles.

This could also be re-written as a mini-review, expanding on the well written discussion section, expanding on the strengths and weaknesses of published data on this subject.

The ethics statement is appropriate.

Reviewer 2 Report

The manuscript is well done and has satisfied all curiosities on the subject. It would be interesting to clarify whether 'tail fragmentation' can influence the robustness of the treatment plan as well as the uncertainty of the dose distribution, especially for plans with multiple beam directions with small targets nearby of the OARs.

Reviewer 3 Report

Referee report on

Rotating gantries provide individualized beam arrangements for charged particle therapy

by Siven Chinniah

General remarks:

The paper describes a retrospective analysis of beam arrangements used to treat 467 patients by proton beams with various curative treatment plans for various tumor sites. In total it is an interesting study on the state of the art of using various beam angles to treat a patient by proton therapy. In general the paper is carefully written and the data analysed in detail. The paper shows what is done at a proton therapy center that has a proton gantry available.

However, there are some points that remains unclear:

1)    Although always stated, that using a gantry implies several advantages, it remains unclear whether this thesis can be underlined with a more quantitative analysis of the advantages. Although there are several “soft” arguments, the thesis may not be proven. The list of used beam angles, as detailed in this paper, may be not sufficient to prove this. In order to prove, treatment plans of different settings may have to be compared, may be on the same subset of patients analysed in this paper and to be examined on outcomes in terms of tumor control and side effects:

-       Treatment planning protons with gantry

-       Treament planning protons with some fixed beams only

-       Treatment planning protons, fixed angles, rotating and tilting patients in chair position and/or supine position

2)    The title of the paper points to the usefulness of gantries in charged particle therapy as a whole. On the other hand the motivation of the project is pointed out to be carbon ion therapy and whether a gantry is needed in that case. The authors argue that they analyse the treatment parameters of patients that obtained proton therapy but where these patients would be eligible for carbon ion treatment. It remains unclear what makes these patients eligible for carbon ion treatment although these patients were treated by proton beams. It remains even unclear why these patients could have been treated with protons and what the patients would gain from carbon ion therapy. In order to do so, the investigations of point 1 would have to be extended to carbon ion treatments with the same parameters.

In total, at least: how many patients suffered from becoming proton therapy but would profit from carbon ion therapy for the reinvestigated cohort of patients? Would the patients profit from carbon ion therapy even if no gantry for the carbon beams would be available.

In that sense the authors should complete information or reduce their claims regarding the necessity of a gantry, in particular in view of carbon ion treatment.

Detailed comments:

There is some confusion in the simple summary that proton data are used to obtain information on carbon ion therapy (lines 22 ff). What has been done in reality and what is done retrospectively in order to assess the necessity in carbon ion treatment. Please clarify.

Reviewer 4 Report

See the instruction in the pdf file.
